# OpenReview forum: "ReSum: Unlocking Long-Horizon Search Intelligence via Context Summarization"
_ICLR.cc/2026/Conference — Submitted to ICLR 2026_

### Official Review · Reviewer_Je2p · 2025-10-29

**Soundness:** 3
**Presentation:** 4
**Contribution:** 2
**Rating:** 4
**Confidence:** 3

**Summary:**

This work focuses on web agents for information-seeking tasks and introduces the ReSum framework to address the context constraint problem that inevitably arises in long-horizon explorations. The framework periodically summarizes ongoing conversations into structured summaries. The summarization process aims to extract key clues and evidence from lengthy interactions, identify information gaps, and highlight next-step directions.
To enable this effectively, this work presents ReSumTool-30B, a summarization model designed for web-context interactions, and ReSum-GRPO, an algorithm that allows the agent to operate more effectively based on the summarized context.

**Strengths:**

- **Potential for small-scale models:** The proposed ReSum framework demonstrates the possibility of building effective web agents even with relatively small base models that have limited context windows, through the mechanism of context compression.

- **Introduction of ReSumTool-30B:** This paper proposes a web-context summarization model that performs comparably well with a relatively modest parameter size (30B), suggesting its efficiency and practicality.

- **Empirical findings from ReSum-GRPO:** The experiments related to ReSum-GRPO show that 1) agents perform significantly better when they learn how to utilize summarized contexts, and 2) ReSum-GRPO serves as a more effective reinforcement learning strategy compared to standard RL approaches.

**Weaknesses:**

While the proposed ReSum framework forms the core of this paper, it raises several conceptual and methodological concerns:

- **Lack of discussion on the distinction from world models:** This paper defines the goal of summarization as extracting key clues and evidence, identifying information gaps, and highlighting next-step directions (lines 79–80). Conceptually, this process 1) converts the interaction history into a structured format more interpretable by LLMs and 2) provides cues for the agent’s subsequent actions. In this sense, **I think ReSum’s summarization mechanism is highly similar to the World Model [1] specialized for information-seeking tasks.** Similarly, world models simulate the next observation from previous trajectories, thereby **offering structured summary and next-step guidance to the LLM.** The primary difference lies in the output format adapted to the specific task, while the underlying methodology remains fundamentally aligned. Consequently, I believe the paper would benefit from **a clearer articulation of the similarities and differences between ReSum and existing world-model approaches,** which is essential to substantiate the claimed novelty of the ReSum framework.

- **Inference cost concerns:** As shown in Figure 8 of Appendix H.2, the ReSum framework incurs an increased inference cost. However, this overhead may undermine the claimed efficiency of achieving strong performance with smaller models. If a larger model (e.g., GPT-OSS in Table 1) achieves higher performance at a lower overall inference cost than the ReSum-enhanced smaller model, **the advantage of ReSum diminishes considerably.** To address this concern, the paper **should include comparative inference-cost analyses across different base models,** highlighting whether the performance gains justify the additional computational expense.

**Questions:**

- Do you believe that the ReSum framework would remain effective for agents built upon base models with sufficiently large context windows? While I acknowledge that ReSum’s primary objective is context compression, prior works [2][3] have shown that converting context into structured representations can improve reasoning and planning even in models with larger context capacity. It would therefore be valuable to examine whether ReSum yields similar benefits under such settings.


- In Table 1, what are the specific differences between the ReAct and Recent History baselines in terms of setup? A more detailed explanation of these configurations would clarify how the comparison was conducted.


- Could you provide additional results comparing inference cost and performance across different base models (e.g., GPT-OSS and others)?

[1] Web Agents with World Models: Learning and Leveraging Environment Dynamics in Web Navigation; Chae et al.; ICLR 2025

[2] Chain-of-Note: Enhancing Robustness in Retrieval-Augmented Language Models ; Yu et al.; EMNLP 2024

[3] Improving Retrieval Augmented Language Model with Self-Reasoning; Xia et al.; AAAI 2025

---

> ### Author Response · Authors · 2025-11-19
> **Response to Reviewer Je2p (1/2)**
>
> We sincerely thank the reviewer for your insightful and constructive feedback.
>
> ---
>
> > W1: Lack of discussion on the distinction from world models
>
> Thank you for the insightful connection to world models. We agree that both approaches utilize an external component to provide structured guidance. However, we respectfully argue that ReSum and world models **address fundamentally different problems** and thus operate on **distinct principles**. The comparison table below summarizes the core differences:
>
> || World Model [1] | ReSum |
> | -------- | -------- | -------- |
> | Problem | Action selection under uncertainty | Context window exhaustion|
> |Primary Goal|Improve decision **quality**|Enable decision **continuation**|
> |Role in Agent|Dense, forward-looking planner|Sparse, backward-looking context manager|
> |Invocation Frequency|High (e.g., 50 steps x 5 samples = 250 calls / query)|Low (~1-3 calls / query)|
>
> We elaborate on the key distinctions:
>
> * **Divergent Goals: Decision Quality vs. Exploration Feasibility**  World models in [1] are introduced to improve the quality of each action by simulating the environment's dynamics. In contrast, ReSum is proposed to solve the problem of premature termination. Without ReSum, many complex queries are **infeasible** due to context overflow, regardless of decision quality. ReSum's goal is to enable this previously impossible exploration.
>
> * **Orthogonal Roles: Planner vs. Context Manager** The world model acts as a planner that is deeply integrated into the reasoning loop at **every step** to evaluate candidate actions. ReSum acts as a context manager that sits outside the core reasoning loop.  It is passively triggered only when a system constraint, e.g., context limit, is breached. The occasional next-step suggestions in summaries are an emergent byproduct, not the design intent.
>
> We have added this clarifying discussion to Appendix A of the revised manuscript.
>
> > W2 & Q3: Inference cost concerns
>
> Thank you for raising this practical concern. We argue that ReSum's performance gains justify its cost from three aspects:
>
> **(1) Favorable cost-performance trade-off:** The added computational cost is the ***necessary enabler*** for solving more complex queries that base agents fail to complete. To provide a direct comparison, we present a comprehensive cost-performance analysis. The results show that ReSum-enhanced 30B models achieve performance competitive with much larger models while requiring significantly less demanding hardware.
>
> | Model & Method | Hardware   | Pass@1 (%) on BrowseComp-zh| # Tokens (K) |
> |-|-|-|-|
> | WebSailor-30B ReAct          | 1x40G GPU  | 23.9                    | 22.9        |
> | WebSailor-30B ReSum-GRPO     | 2x40G GPU  | 33.3                    | 41.9        |
> | Tongyi-DeepResearch-30B ReAct | 1x40G GPU  | 43.6                    | 44.7        |
> | Tongyi-DeepResearch-30B ReSum | 2x40G GPU  | 48.6                    | 76.1        |
> | GPT-OSS-120B ReAct         | 1x144G GPU | 38.0                    | 28.3        |
> | DeepSeek-V3.1-671B  ReAct | 8x144G GPU | 49.2                    | 40.6        |
>
> **(2) Addressing deployment-constraint scenarios:** The most compelling use case for ReSum is in resource-constrained environments, e.g., on-device deployment, cost-sensitive applications, where 144G+ GPUs are unavailable. In these settings, the relevant question is not "ReSum vs. GPT-OSS", but rather "**What is the best performance achievable with 80G of GPU memory?**" ReSum provides a clear path to boost the capabilities of smaller, more deployable models, offering a practical solution for real-world applications.
>
> **(3) Orthogonal to model capability:** As shown in the table, ReSum also substantially improves the stronger Tongyi-DeepResearch-30B backbone. This demonstrates that the paradigm offers gains that are orthogonal to base agent capability.
>
> ---
>
> [1] Chae et al. *Web Agents with World Models: Learning and Leveraging Environment Dynamics in Web Navigation.* In ICLR, 2025

---

> ### Author Response · Authors · 2025-11-19
> **Response to Reviewer Je2p (2/2)**
>
> > Q1: ReSum applied on models with large context window
>
> Thank you for this valuable question. We conducted experiments on Tongyi-DeepResearch-30B-A3B, a state-of-the-art open-source web agent with extensive context window.
>
> We progressively increased the context window from 32K to 128K. For each setting, ReAct terminates upon hitting the context limit, while ReSum compresses the history to continue exploration. To ensure a fair comparison, the maximum tool call limit was set proportionally to each context size (e.g., 40 calls for 32K, 120 calls for 128K). As shown below, **ReSum consistently outperforms ReAct even with extensive context capacity**:
>
> *Format: ReAct → ReSum (Δ absolute improvement)*
>
> | **Context Limit** | **BrowseComp-zh Pass@1** | **BrowseComp-zh Pass@3** | **BrowseComp-en Pass@1** | **BrowseComp-en Pass@3** |
> |-------------------|--------------------------|--------------------------|--------------------------|--------------------------|
> | 32K (40 calls)    | 41.2 → 43.8 (+2.6)   | 57.4 → 62.3 (+4.9)   | 27.7 → 34.5 (+6.8)   | 43.1 → 53.3 (+10.2)  |
> | 48K (60 calls)    | 42.5 → 46.7 (+4.2)   | 58.5 → 62.6 (+4.1)   | 32.8 → 38.2 (+5.4)   | 48.7 → 54.5 (+5.8)   |
> | 64K (80 calls)    | 43.6 → 48.6 (+5.0)   | 60.9 → 66.1 (+5.2)   | 36.3 → 40.3 (+4.0)   | 52.4 → 57.8 (+5.4)   |
> | 96K (100 calls)   | 46.0 → 47.9 (+1.9)   | 62.3 → 66.4 (+4.1)   | 39.8 → 41.0 (+1.2)   | 56.5 → 57.0 (+0.5)   |
> | 128K (120 calls)  | 45.7 → 46.6 (+0.9)   | 62.3 → 62.6 (+0.3)   | 42.2 → 44.5 (+2.3)   | 59.2 → 59.5 (+0.3)   |
>
> These results demonstrate that the ReSum paradigm provides a consistent and effective mechanism for extending an agent's exploration horizon, delivering performance gains across a wide range of context constraints. We have added this analysis to Appendix F of the revised manuscript.
>
> > Q2: Configurations between ReAct and Recent History
>
> Thank you for the opportunity to clarify. As detailed in **Appendix D.1**, the experimental configurations are as follows:
>
> * ReAct: Appends every thought, action, and observation to the conversation history. The agent terminates as a failure if it reaches the context window limit before generating an answer.
> * Recent History: Upon reaching the context limit, truncates the conversation history by preserving only the most recent *22K* tokens, allowing continued exploration while maintaining space for new interactions.
>
> To ensure fair comparison, all methods (ReAct, Recent History, ReSum) use a consistent maximum of 60 tool calls and identical 32K context limits.
>
> ---
>
> *Once again, we sincerely thank you for your constructive feedback and suggestions. We hope the additional experiments and clarifications address the raised points and further substantiate ReSum's contribution. Please let us know if there are any further questions we can answer.*

---

> ### Author Response · Authors · 2025-11-24
> **Thank you & Looking forward to further discussion**
>
> Dear Reviewer Je2p,
>
> We sincerely thank you for your time and constructive feedback throughout the author–reviewer discussion process. To help facilitate your review of our responses, we would like to briefly highlight how we have addressed your primary concerns:
>
> * **Distinction from World Models** `Weakness1` We have clarified that ReSum and world models address fundamentally different problems under distinct principles. ReSum focuses on enabling continued exploration by serving as a sparse, backward-looking context manager, while world models aim to improve decision quality by acting as a dense, forward-looking planner.
> * **Effectiveness on models with large context windows** `Question1` We have added experiments showing that ReSum provides consistent performance gains even for agents with extensive context capacity, i.e., up to 128K.
> * **Efficiency justification for ReSum** `Weakness2, Question2` We have supplemented the cost–performance analysis, demonstrating that ReSum offers a lightweight and deployable alternative to very large models, while also providing orthogonal performance gains across diverse agent capabilities.
>
> For other issues not mentioned here, please refer to our detailed rebuttal response. We sincerely hope this addresses your concerns and respectfully look forward to further discussion with you.
>
> Warm regards,
>
> Authors of submission 8387

---

> ### Author Response · Authors · 2025-11-26
> **[Only 1 Week Remaining]  A Gentle Reminder for Feedback**
>
> Dear Reviewer  Je2p,
>
> We sincerely thank you for your valuable feedback and the time you have invested in reviewing our submission. We have carefully addressed the concerns you raised, incorporated relevant discussions and experiments, and revised the manuscript accordingly.
>
> As the deadline for discussion is approaching, we kindly request your feedback to confirm **whether our responses sufficiently address your concerns**. If there are any remaining issues, we would greatly appreciate the opportunity to address them to further improve the quality of our work.
>
> Thank you again for your time and consideration.
>
> Warm regards,
>
> Authors of submission 8387

---

### Official Review · Reviewer_FmG8 · 2025-10-31

**Soundness:** 2
**Presentation:** 2
**Contribution:** 2
**Rating:** 4
**Confidence:** 4

**Summary:**

This paper introduces ReSum, a simple extension of ReAct for reasoning-then-act agents on long-horizon agentic tasks. In ReSum, the history of agent reasoning, action, and observations in previous iterations are periodically summarized into key takeaways and information gaps to recombine with the original query prompt, to avoid the agent history from running out of the context window of typical LLMs.

The paper also presents ReSumTool-30B, a smaller, specialized summarization model distilled from larger models that helps ReSum achieve good performance without the inference cost that comes with larger models. Furthermore, when GRPO is applied to ReSum trajectories (ReSum-GRPO), empirical results show further gains in agent performance.

**Strengths:**

1. The paper presents a relatively simple yet effective solution to the problem of ReAct agent context overflow by summerizing past agent reasoning, actions, and observations in the agent history. The solution is well-motivated and clearly explained.
2. The paper presents extensive experimentation with empirical results that demonstrates ReSum achieving gains over ReAct, as well as simpler context-fitting adaptations of ReAct such as agents that truncate distant history to only retain the most recent reasoning, actions, and observations ("Recent History").

**Weaknesses:**

1. The paper makes unsubstantiated or inaccurate claims about the proposed approach in a few places. For instance,

    1. On L261, "Unlike most agentic RL approaches (Liu et al., 2025; Dong et al., 2025) that impose format rewards, ours relies solely on answer correctness to provide a more result-oriented signal. Additionally, we perform format checks at each generation step: if the agent fails to adhere to specific tokens such as <think> </think>, the entire trajectory is terminated and assigned a zero reward as a penalty. This implicitly guides the agent to follow the required format effectively." What the stated approach is doing is precisely a form of format rewards, just instead of an additive reward, the format reward imposes a nullifying penalty to prevent any partial credits unless formatting requirements are met.
    2. On L291, one of the stated benefits of ReSum-GRPO is "strategic information gathering to collect evidence that yields high-quality summaries". This has not been empirically verified by experiments.
    3. On L366, the paper states "ReSum paradigm consistently outperforms ReAct due to extended exploration opportunities." This is not substantiated by comparing success rate of long trajectories vs shorter ones, for instance.

2. Questionable evaluation and comparison.

    1. The main evaluation performed in the paper uses LLM as a judge (L310: "We consistently use Qwen2.5-72B-Instruct as the scoring model to assess whether the predicted answer aligns with the ground truth."), but there has never been any evaluation of this evaluation method itself to make sure it is correct. Some benchmarks, take GAIA, for instance, come with official evaluation functions. It is unclear why the paper adopts an ad hoc evaluation metric when official ones are available for better comparability across the literature.
    2. The only comparison to prior approaches of intelligent agent context compaction is in Appendix F, and only one prior work (MEM1) is compared. For a better evaluation of the contributions of the proposed approach, this should have been more comprehensive and be part of the main body of the paper.
    3. In GRPO experiments, the paper leaves out "Recent History" as a potential compaction approach.
    4. The paper shows ReAct performance of many leading closed- and open-weights models, but never apply ReSum to these agents. Is it because ReSum is not effective when the base model is strong enough? (which would also be useful to know)

3. Contradicting statements and misrepresented results.

    1. On Line 725, the paper states "This design (ReSum) preserves the simplicity and efficiency of ReAct while ensuring seamless compatibility with existing agents.", while on Line 236 it concedes "This pattern (of summarized history) is out-of-distribution for standard agents, as they have not encountered summary-conditioned reasoning during training." While empirically we see that LLMs are able to make use of the summarized history in most cases, one of these statements is less accurate than the other.
    2. On Line 399, the paper states "ReSum integration effectively narrows the performance gap to leading pre-trained models." This conveniently leaves out the significantly superior performance of OpenAI o3 and DeepSeek v3.1, and also does not answer the question of where the performance gap might be.
    3. In Table 1, the authors use bold fonts to indicate results from the proposed approach rather than the conventional best results (even if within the same model size class), which is potentially misleading to the casual reader.

**Questions:**

1. What's the objective of the summarization, is there any evaluation of summary quality aside from prompt specification of the behavior of the teacher model itself?
2. L253 states "Each segment H(i) forms an individual training episode with input q(i−1) and output ($\tau_{t-1}$, ...$a_{t_i}$)." How is the loss calculated, is the model also updated to predict the $o_t$'s?
3. What is the variance of the performance of the agent across independent reruns? Are the differences between the systems statistically significant / meaningful?

---

> ### Author Response · Authors · 2025-11-19
> **Response to Reviewer FmG8 (Part 1/3)**
>
> We sincerely thank the reviewer for your thorough and constructive feedback.
>
> ---
>
> > W1: Unsubstantiated or inaccurate claims
>
> Thank you for your careful reading. We clarify these points below:
>
> * **Format rewards (L261)** We acknowledge that our format-checking mechanism is indeed a form of format reward, i.e., a binary, nullifying penalty. Our intent was to distinguish it from approaches that assign *continuous, additive rewards* for partial formatting compliance. We have revised the description accordingly to avoid confusion.
>
> * **Strategic information gathering (L291)** While our reward broadcasting mechanism incentivizes agents to collect information that facilitates high-quality summaries (since all segments receive the trajectory-level advantage), quantitatively measuring this poses challenges, as it requires **defining ground-truth evidence sets for each open-ended query**, which is difficult to establish. Instead, we provide qualitative example in **Appendix G.3 (Case 2)** showing agent behaviors that suggest strategic information gathering.  Finally, to prevent potential misunderstanding, we have revised the original phrasing to more precisely reflect the mechanism as
> *ensuring early exploration steps receive appropriate bonus when they contribute to the final success*.
>
> * **Extended exploration opportunities (L366)** We support our claim from two aspects:
>    * Under the same tool calling budget, ReSum achieves a higher success rate than ReAct. This demonstrates its ability to **complete trajectories that ReAct cannot**, due to context-exhausted premature termination.
>    * ReSum also outperforms other unbounded baselines like MEM1, and Recent History. This confirms the gain stems from effective context summarization, not merely from having a longer history. It is the *quality* of the extended exploration, that drives the improvement.
>
> > W2.1 Justification for LLM-as-Judge
>
> We appreciate this concern about evaluation rigor and provide our rationale below:
>
> * **Alignment with existing literature** Our use of Qwen2.5-72B-Instruct for judgment follows standard practice in recent web agent research, including ASearcher [Ref A] and ARPO [Ref B], facilitating direct comparability within this emerging field.
>
> * **Comparison of judging models** We compared 4 judging models, Qwen2.5-72B-Instruct, GPT-4o-Mini, Qwen3-235B, and Gemini-2.5-Flash, on WebSailor-7B predictions for BrowseComp-zh. The resulting scores were 12.8%, 12.5%, 12.8%, and 12.5% respectively, showing variance <0.3% across all models. Discrepancies arose primarily from **semantic differences in answer expression**. For example, when the ground truth was an English paper title and the model predicted its Chinese translation, Qwen2.5 and Qwen3 correctly identified equivalence while others did not. Therefore, we selected Qwen2.5-72B-Instruct for its **reliability and lower cost**.
>
> * **Consistency with official evaluation protocols:** For each benchmark, we adopt the same judging prompts from official evaluations, only replacing the judging model. This ensures our evaluation aligns with official benchmarks.
>
> > W2.2 Choices of baselines
>
> Thank you for your suggestion of moving the MEM1 comparison to the main text, which we have followed by relocating it to Section 4.4.  We also clarify our baseline selection rationale:
>
> * Limited prior work on context management for web agents: Prior to our work, context management for web agents was largely unexplored. Leading models, WebSailor [Ref C], ASearcher, and WebExplorer [Ref D] consistently adopt the standard ReAct paradigm without any context curation mechanisms.
>
> * Existing agent memory mechanisms are not directly applicable: While memory mechanisms exist for agents, most focus on external knowledge injection, i.e., incorporating information from outside the current task, such as past trajectories or experiences across different queries. Our work addresses a different challenge: **managing the continuously accumulating context** within a single query as agents perform information gathering and persistent search.
>
> From this perspective, MEM1 is both applicable and highly representative, it specifically addresses context compression within long-horizon tasks, making it the most suitable baseline. Moreover, **our evaluation of MEM1 is comprehensive**: we compare across multiple benchmarks (GAIA, BrowseComp-zh, BrowseComp-en) under both training-free and training-required settings, providing thorough empirical analysis of ReSum's advantages.
>
> ---
>
> [Ref A] Gao et al. *Beyond Ten Turns: Unlocking Long-Horizon Agentic Search with Large-Scale Asynchronous RL.*  In arXiv preprint, 2025
>
> [Ref B] Dong et al. *Agentic Reinforced Policy Optimization.* In arXiv preprint, 2025
>
> [Ref C] Li et al. *WebSailor: Navigating Super-human Reasoning for Web Agent.* In arXiv preprint, 2025
>
> [Ref D] Liu et al. *WebExplorer: Explore and Evolve for Training Long-Horizon Web Agents.* In arXiv preprint, 2025

---

> ### Author Response · Authors · 2025-11-19
> **Response to Reviewer FmG8 (Part 2/3)**
>
> > W2.3 GRPO experiments without Recent History
>
> Thank you for your feedback. The primary goal of our GRPO experiments was to validate ReSum-GRPO vs. standard GRPO for ReSum paradigm adaptation.
>
> Applying RL to Recent History would require addressing distinct challenges: its sliding window creates variable context states across generation steps, complicating trajectory segmentation and reward assignment. We believe it requires dedicated algorithmic design beyond this paper's scope. Besides, Recent History showed modest performance gains in the training-free setting, making it a lower-priority candidate for RL investigation.
>
> > W2.4 Experiments on stronger models
>
> Thank you for this important question. We have indeed evaluated ReSum on stronger models, Tongyi-DeepResearch-30B-A3B, the leading open-source web agent at submission time (detailed in **Appendix F**). Our tests under varying context limits demonstrate that **ReSum remains effective even for powerful models with extensive context capabilities:**
> |**Context Limit**|**Paradigm**|**BrowseComp-zh Pass@1**|**BrowseComp-zh Pass@3**|**BrowseComp-en Pass@1**|**BrowseComp-en Pass@3**|
> |---|---|---|---|---|---|
> |64K|ReAct|43.6|60.9|36.3|52.4|
> |64K|ReSum|**48.6(+5.0)**|**66.1(+5.2)**|**40.3(+4.0)**|**57.8(+5.4)**|
> |128K|ReAct|45.7|62.3|42.2|59.2|
> |128K|ReSum|**46.6(+0.9)**|**62.6(+0.3)**|**44.5(+2.3)**|**59.5(+0.3)**|
>
> We kindly invite the reviewer to refer to Appendix F in the revised manuscript, where we have included additional context limit settings. The results consistently demonstrate the effectiveness of ReSum.
>
> > W3: Contradicting statements and misrepresented results.
>
> We thank the reviewer for the meticulous reading, and we have revised the manuscript to improve clarity.
>
> * **Seamless compatibility vs. out-of-distribution** Thank you for the opportunity to clarify. "Seamless compatibility" refers to *functional plug-and-play integration*, where any ReAct-trained agent can adopt the ReSum paradigm during inference without  architectural changes, as the main interaction loop remains unchanged. "Out-of-distribution" refers to *suboptimal performance* when using ReSum without specialization, as this capability was not part of the agent's training data.
>
> * **Narrows the performance gap** We acknowledge that this statement should be more precise. ReSum narrows the performance gap to leading commercial models including Kimi-K2 and Claude-4, while still trailing frontier models like o3 and DeepSeek-V3.1. This gap is reasonable given that these frontier models represent SOTA capabilities built with substantially larger resources in compute, data, and engineering.
>
> * **Table 1** We have revised Table 1 to highlight both our results and the best results in each category to avoid potential confusion. Thank you for this valuable suggestion.
>
> > Q1: Summarization quality
>
> The goal of summarization is to condense lengthy interaction history for continued reasoning while preserving essential information. We evaluate summary quality in **Appendix E** through both qualitative analysis and downstream performance metrics.
>
> * **Qualitative analysis** We compare summaries generated by GPT-OSS-120B and Qwen2.5-72B-Instruct. GPT-OSS-120B produces higher-quality summaries with structural organization, comprehensive evidence gathering, and goal-oriented focus. We distill this capability into ReSumTool-30B, and Figure 6 demonstrates that it successfully achieves comparable summarization quality.
> * **Downstream performance evaluation** Since agents rely on summaries to resume exploration, downstream task performance serves as a quality metric. As shown in Table 1, agents using ReSumTool-30B achieve strong performance, significantly outperforming agents using base Qwen3-30B, demonstrating ReSumTool's effectiveness.
>
> > W2: Loss computation
>
> As stated in **Appendix D.2**, loss computation only considers agent-generated content: thoughts $\tau$ and tool invocations $a$. All observations $o$ are masked during loss calculation.

---

> ### Author Response · Authors · 2025-11-20
> **Response to Reviewer FmG8 (Part 3/3)**
>
> > W3: Model variance
>
> Thank you for raising this concern. Following standard practice in recent literature, we report Pass@1 (average success rate across three independent runs) and Pass@3 (success within three attempts). Below we provide standard deviations of **Pass@1** across three runs, showing **variance <2% in most cases**. Given these low variances, the observed improvements, **+4.5% for training-free ReSum and +8.2% for ReSum-GRPO, are statistically meaningful and demonstrate consistent gains across runs**.
>
> WebSailor-3B
> |Setting|GAIA|BrowseComp-zh|BrowseComp-en|
> |-|-|-|-|
> |ReAct|25.6±2.3|8.2±0.7|3.3±1.1|
> |ReSum|35.3±5.7|13.7±1.3|6.8±1.9|
> |GRPO-ReAct|28.5±3.1|11.8±0.6|4.2±0.7|
> |GRPO-ReSum|38.5±3.7|17.3±1.8|8.5±1.3|
> |ReSum-GRPO|37.9±1.7|20.5±0.5|9.2±0.3|
>
> WebSailor-7B
> |Setting|GAIA|BrowseComp-zh|BrowseComp-en|
> |-|-|-|-|
> |ReAct|31.7±1.1|13.2±0.5|5.7±1.1|
> |ReSum|40.5±1.1|17.2±3.4|9.0±0.4|
> |GRPO-ReAct|34.0±0.9|18.7±2.7|5.8±1.1|
> |GRPO-ReSum|37.2±1.4|25.4±0.5|8.5±0.4|
> |ReSum-GRPO|42.4±2.0|27.1±0.5|12.3±0.6|
>
> WebSailor-30B
> |Setting|GAIA|BrowseComp-zh|BrowseComp-en|
> |-|-|-|-|
> |ReAct|45.0±3.9|23.9±2.1|12.8±1.5|
> |ReSum|47.3±1.9|24.1±1.9|16.0±1.7|
> |GRPO-ReAct|48.2±2.0|23.3±0.8|14.3±1.1|
> |GRPO-ReSum|48.5±1.7|29.3±1.1|15.0±1.7|
> |ReSum-GRPO|48.5±3.3|33.3±1.7|18.3±1.5|
>
> ---
>
> *Once again, we sincerely thank you for your meticulous reading, constructive feedback, and valuable suggestions. We hope the additional experiments and clarifications in the revised manuscript have addressed the points you raised. Please let us know if there are any further questions we can answer.*

---

> ### Author Response · Authors · 2025-11-24
> **Thank you & Looking forward to further discussion**
>
> Dear Reviewer FmG8,
>
> We extend our highest respect for the time and effort you devoted to reviewing our manuscript. To facilitate better communication and express our gratitude, we sincerely summarize our rebuttal as follows:
>
> * **LLM-as-Judge** `Weakness2.1` We have justified our use of Qwen2.5-72B-Instruct based on its alignment with established practices in web agent research, minor variance across multiple judge models, and strict adherence to official evaluation prompts for each benchmark.
>
> * **Performance variance** `Question3` We have supplemented the main results with standard deviations across three independent runs, showing variances <2% in most cases, confirming that the performance improvements from ReSum are meaningful.
>
> * **Effectiveness on stronger models** `Weakness2.4` We have added experiments on Tongyi-DeepResearch-30B-A3B, a leading open-source agent, demonstrating that ReSum provides performance gains even on more capable base models.
>
> In addition, we have carefully revised the manuscript to improve accuracy in descriptions (`Weakness1`) and table presentation (`Weakness3`) to avoid any potential misunderstanding. For any other issues not mentioned here, please refer to our detailed rebuttal response. We fully and humbly understand that you might have a tight schedule, and we would be sincerely grateful if you could let us know **whether our rebuttal has sufficiently addressed your concerns**.
>
> Warm regards,
>
> Authors of submission 8387

---

> ### Author Response · Authors · 2025-11-26
> **[Only 1 Week Remaining] A Gentle Reminder for Feedback**
>
> Dear Reviewer  FmG8,
>
> We sincerely thank you for your valuable feedback and the time you have invested in reviewing our submission. We have carefully addressed the concerns you raised, incorporated relevant discussions and experiments, and revised the manuscript accordingly.
>
> As the deadline for discussion is approaching, we kindly request your feedback to confirm **whether our responses sufficiently address your concerns**. If there are any remaining issues, we would greatly appreciate the opportunity to address them to further improve the quality of our work.
>
> Thank you again for your time and consideration.
>
> Warm regards,
>
> Authors of submission 8387

---

> ### Comment · Reviewer_FmG8 · 2025-11-27
> **Thank you for the response.**
>
> I appreciate the authors for their clarification in the response.
>
> In the response, the authors state
> > Consistency with official evaluation protocols: For each benchmark, we adopt the same judging prompts from official evaluations, only replacing the judging model. This ensures our evaluation aligns with official benchmarks.
>
> This is far from sufficient to ensure comparability, and does not address my original comment about not adopting the official evaluation methods provided by some of the benchmarks (e.g., GAIA's evaluation is based on exact matches with the ground truth answer). This continues to undermine the comparability of the results presented in the manuscript to previous work.
>
> I also appreciate the authors providing model variance results. As can be seen in the results in response 3/3, while the gains on smaller models (3B and 7B) are more salient, the gains on the 30B model is statistically indistinguishable with overlapping confidence intervals in most places. This undermines the argument that the proposed approach is also effective for more powerful models -- and should have been acknowledged.

---

> > ### Author Response · Authors · 2025-11-27
> > **Further Clarification on GAIA Evaluation & Improvement**
> >
> > Thank you for your valuable feedback and the opportunity to further clarify.
> >
> > * **GAIA Evaluation** To ensure a **consistent and fair agent inference process** across all benchmarks, we adopted the established LLM-as-Judge protocol used in existing web agent literature (e.g., WebSailor, ASearcher, WebExplorer). This choice is further validated by our preliminary experiments, which show that the judgments exhibit very low variance and achieve high agreement with human evaluators. The official GAIA protocol relies on a task-specific system prompt to format model outputs, a step that differs from the standard workflow of general web agents. Using a unified evaluation method allows our results to be **directly comparable to contemporary web agents without introducing benchmark-specific adaptations**.
> >
> > * **Improvement across models** We observe consistent improvements across all model sizes, demonstrating the broad applicability of ReSum.
> >    *  For WebSailor-30B, performance on BrowseComp rises substantially, from 23.9 to 33.3 on BrowseComp-zh (+9.4) and from 12.8 to 18.3 on BrowseComp-en (+5.5).
> >    * The more pronounced gains in smaller models underscore ReSum's **practical value** in resource-limited or on-device deployment, improving the utility of compact models.
> >    * As shown in [Response 2/3], ReSum also **boosts performance on leading web agents**, i.e., Tongyi-DeepResearch-30B-A3B, supporting its general effectiveness.
> >
> > ---
> >
> > We sincerely appreciate your insightful comments and hope these clarifications fully address your concerns.

---

### Official Review · Reviewer_oXox · 2025-10-31

**Soundness:** 3
**Presentation:** 3
**Contribution:** 2
**Rating:** 4
**Confidence:** 3

**Summary:**

In this paper, the authors are trying to address a critical challenge of LLM-based agents with paradigms like ReAct, i.e., the fixed context window. The growing history of thoughts, actions, and observations quickly exhausts the available context, forcing premature termination before a solution is found. To solve this problem, in this paper, a method called ReSum is proposed, which enables indefinite exploration through periodic context summarization. A summarization tool ReSumTool-30B is proposed to extract key evidence and identify critical information gaps from lengthy interactions. The paper further presents the ReSum-GRPO reinforcement learning algorithm to adapt agents to summary-conditioned reasoning and demonstrate, across three established benchmarks, superior performance over ReAct.

**Strengths:**

1. The paper is well written with good structure and is easy to follow. It presents a clear, practical, and effective solution to a significant and practical challenge in agentic AI.

2. The ReSum-GRPO algorithm is a novel adaptation of RL. The concepts of trajectory segmentation and advantage broadcasting are well-suited for this specific problem and avoid the high cost of expert-annotated SFT data. addresses the distribution shift of summary-conditioned inputs by segmenting rollouts and broadcasting trajectory-level advantages, a reasonable and well-motivated modification that preserves GRPO’s structure while handling long trajectories.

3. The proposed framework achieves good performance over other baselines. The proposed method is evaluated across multiple agent scales, benchmarks, and both training-free and RL settings. The authors also provide resource/cost vs. performance analyses and case studies showing qualitative behaviour. This breadth strengthens the empirical claim.

**Weaknesses:**

1. In my opinion, the triggering strategy is rule-based and limited by the "naive" trigger, e.g., approaching the token budget. This is simple but may not be optimal, i.e., may encounter the case that is in the middle of a complex chain?

2. As the proposed method follows a periodic context summarization and the entire paradigm depends on the quality of this summary, I wonder if this will include the risk of error propagation. For example, some crucial information can be lost and making the task really hard to solve.

3. Although the paper effectively analyzes total token consumption and training time, it does not fully address the inference-time latency of the summarization step itself. Each call to ReSumTool-30B is an additional LLM call that pauses the agent's reasoning. For a long task requiring many summaries, this could add significant latency. A more detailed analysis of this latency would be valuable.

4. The paper provides a high performance summary model, ReSumTool-30B. This seems like a big model for this task. However, only 1K training samples can make it effective. I wonder if the balance of these setups could be further studied.

**Questions:**

Please check the questions in the Weaknesses, and also address the following questions.

1. I wonder if more triggering strategies can be included for testing. Also, if the trigger frequency changes, will the performance also change a lot?

2. Since summarization can abstract away details, how does the agent mitigate the risk of introducing inaccuracies or hallucinations in ongoing reasoning?

---

> ### Author Response · Authors · 2025-11-19
> **Response to Reviewer oXox (Part 1/2)**
>
> We sincerely thank the reviewer for your insightful and constructive feedback.
>
> ---
>
> > W1: Trigger for summarization
>
> Thank you for this insightful feedback. We acknowledge that triggering summarization could potentially disrupt complex reasoning chains. However, our context-limit trigger **minimizes this risk** as summarization occurs only when context would otherwise overflow, allowing maximum uninterrupted reasoning. Besides, such rule-based approach is simple to implement and ensures plug-and-play compatibility.
>
> We agree that **learnable triggers**, where agents autonomously decide to manage their context through summarization, would be more elegant. However, this requires cognitive capabilities that exceed current agents' scope, which we leave as important future work.
>
> > W2 & Q2: Risk of error propagation
>
> Thank you for raising this important concern. We recognize that summary quality directly impacts ReSum's effectiveness. Therefore, we address this from two aspects:
>
> * **Rigorous summary tool quality control (Appendix E)** We conduct systematic quality control for ReSumTool-30B through:
>    - Teacher model selection: We compare leading LLMs and select GPT-OSS-120B for its ability to intelligently extract essential information from lengthy, noisy interactions and structure it with clear organization and source attribution
>    - Extensive data collection: We collect 9K+ high-quality `<Conversation, Summary>` pairs to distill this capability into a smaller-scale model
>    - Comprehensive evaluation: We perform in-depth case analysis on the distilled ReSumTool-30B to ensure reliability
>
>     Empirically, Table 1 demonstrates that ReSumTool-30B maintains strong performance across benchmarks, validating the effectiveness of our quality control pipeline.
>
> * **Persistent agent behavior via ReSum-GRPO** Additionally, our RL algorithm uses **answer correctness as the reward signal**, which inherently encourages persistent exploration. Even when a summary omits information, the trained agent actively continues searching and browsing until the query is solved.
>
>
> > W3: Inference latency
>
> Thank you for raising this practical concern. Taking WebSailor-30B after ReSum-GRPO training as an example, the average invocation of ReSumTool-30B is **1.4** on BrowseComp-zh and **2.2** on BrowseComp-en. This corresponds to only 1-3 additional LLM calls per query, which introduces manageable overhead. We emphasize that this additional cost is **modest compared to the full trajectory**, which requires an average of **10+ tool calls** (search, browse, extract) and corresponding agent generation steps. Additionally, ReSum enables solving queries that would otherwise fail due to context overflow. The incremental latency is thus an acceptable trade-off for enabling indefinite exploration.
>
> > W4: ReSumTool-30B training data
>
> We respectfully clarify that there may be a misunderstanding regarding the training data volumes:
>
> - **ReSumTool-30B (the summary model)**: Trained with 9K+ high-quality `<Conversation, Summary>` pairs to specialize in evidence extraction and structured summarization (Appendix E)
> - **ReSum-GRPO (agent policy training)**: We leverage 1K `<Question, Answer>` pairs for ReSum-GRPO training to adapt agents to the ReSum paradigm (Sec 4.3)
>
> Regarding the model size, ReSumTool-30B is based on an **MoE architecture** that requires **less memory** than dense models of comparable scale. Additionally, the task of summarizing lengthy, multi-step web interactions demands strong reasoning and a large context window, for which smaller models are inherently less capable.
>
> For quality control of the summary model, we ensure reliability through **rigorous teacher model selection**, **extensive high-quality data collection** (9K+), and **comprehensive case analysis** (Appendix E).

---

> ### Author Response · Authors · 2025-11-19
> **Response to Reviewer oXox (Part 2/2)**
>
> > Q1: Summarization trigger
>
> Thank you for this valuable suggestion. We have conducted additional experiments analyzing different trigger frequencies (8K, 16K, 24K, 32K tokens), where summarization is invoked whenever the context approaches each threshold. All experiments use a consistent maximum of 60 tool calls with WebSailor-7B on GAIA and BrowseComp-zh in the training-free setting.
>
> |Summary Trigger|Pass@1 GAIA (%)|# Summary|# Token (K)|# Tool|
> |---------------|---------------|------------|-----------|------|
> |ReAct|31.7|0|17.4|10.3|
> |8K|44.5|2.6|36.0|17.2|
> |16K|40.8|1.1|29.4|13.6|
> |24K|39.8|0.4|20.7|11.2|
> |32K|40.5|0.3|21.8|12.9|
>
> |Summary Trigger|Pass@1 BrowseComp-zh (%)|# Summary|# Token (K)|# Tool|
> |---------------|----------------|------------|-----------|------|
> |ReAct|13.2|0|29.2|10.7|
> |8K|18.8|5.5|84.5|33.6|
> |16K|21.1|3.7|60.5|21.1|
> |24K|21.5|2.2|58.4|20.4|
> |32K|17.2|1.8|50.6|17.6|
>
> Based on these results, we have the following findings:
>
> * **Consistent benefits of ReSum** All trigger frequencies outperform ReAct, confirming that summarization provides gains through both extended exploration and evidence distillation.
>
> * **Dataset-specific optimal frequency** Simpler benchmarks like GAIA benefit from frequent summarization, e.g., 8K, while more challenging benchmarks like BrowseComp-zh require longer uninterrupted chains as optimal at 24K.
>
> * **Cost-performance trade-off** More frequent summarization significantly increases computational costs due to additional summary invocations, higher token consumption, and an increased number of tool calls. We empirically found that under frequent triggers, e.g., 8K, providing fresh summaries more often stimulates the agent to engage in more persistent verification and exploratory search, leading to a cascade of subsequent actions. Thus, our current implementation, based on the **actual context limit of 32K**, was chosen to **balance between performance gains and practical efficiency**.
>
> ---
>
> *Once again, we sincerely thank you for your constructive feedback and suggestions. We hope the additional experiments and clarifications address the raised points and further substantiate ReSum's contribution. Please let us know if there are any further questions we can answer.*

---

> ### Author Response · Authors · 2025-11-24
> **Thank you & Looking forward to further discussion**
>
> Dear Reviewer oXox,
>
> We sincerely thank you for your thorough review and valuable feedback on our work. To assist in your further assessment, we would like to briefly summarize how we have addressed your primary concerns:
>
> * **Summary quality & error propagation** `Weakness2, Question2` We implemented a rigorous quality control pipeline for ReSumTool-30B (Appendix E), involving a powerful teacher model, extensive data distillation, and in-depth case analysis to ensure reliability and minimize information loss risks.
>
> * **Inference latency** `Weakness3` We have added a latency analysis showing that ReSumTool is invoked only 1–3 times per query on average. This overhead is modest compared to the full trajectory (typically 10+ tool calls and LLM response generation) and is a worthwhile trade-off for enabling the completion of otherwise infeasible tasks.
>
> * **Trigger strategy & frequency** `Weakness1, Question1` We have supplemented an ablation study on trigger frequency. Results show that ReSum brings consistent gains across settings, while our chosen context-limit trigger offers a practical balance between performance and computational cost.
>
> For any other issues not mentioned here, please refer to our detailed rebuttal response. We understand you have a busy schedule and would be truly grateful if you could let us know **whether our responses have adequately addressed your concerns**. Thank you again for your time and insightful comments.
>
> Warm regards,
>
> Authors of submission 8387

---

> ### Author Response · Authors · 2025-11-26
> **[Only 1 Week Remaining] A Gentle Reminder for Feedback**
>
> Dear Reviewer  oXox,
>
> We sincerely thank you for your valuable feedback and the time you have invested in reviewing our submission. We have carefully addressed the concerns you raised, incorporated relevant discussions and experiments, and revised the manuscript accordingly.
>
> As the deadline for discussion is approaching, we kindly request your feedback to confirm **whether our responses sufficiently address your concerns**. If there are any remaining issues, we would greatly appreciate the opportunity to address them to further improve the quality of our work.
>
> Thank you again for your time and consideration.
>
> Warm regards,
>
> Authors of submission 8387

---

### Official Review · Reviewer_SSow · 2025-11-01

**Soundness:** 3
**Presentation:** 3
**Contribution:** 2
**Rating:** 4
**Confidence:** 4

**Summary:**

Authors propose ReSum; a framework that summarizes interaction history; proposing to train their own summarizer, then train an LLM using a modified variant of GRPO to be use to this new format of summary and then further interaction history, as opposed to standard ReACT. Experiments on three web search benchmarks (GAIA, BrowseComp-zh, BrowseComp-en) using WebSailor models (3B–30B) show that ReSum improves Pass@1 by ~4.5% (absolute) over standard ReAct, and applying ReSum-GRPO training yields a further ~8% gain.

**Strengths:**

* This paper tackles an important problem that is highly relevant to the ML community, of how to leverage summaries when executing long-horizon LLM agent tasks.
* Empirical performance of ReSum suggests such an approach is superior compared to ReAct; however the results lack error bars to say anything conclusive.
* Paper is clearly written and easy to follow.
* Plug-and-play: the change to ReAct is minimal (periodic summarization and resume), which makes it easy to adopt across agents.
* Results are reported on three relevant web-agent benchmarks with consistent tooling; inclusion of a MEM1 reproduction is useful context.

**Weaknesses:**

* Improvement of 4.5% over ReAct, could still be within the error bars. I encourage the authors to re-run their experiments for multiple random seeds and quote their error bars.
* Figure 4; statement of “ReSum-GRPO demonstrates higher initial rewards and faster convergence compared to standard GRPO.”, appears to be an overclaim as sub figure b, WebSailor-30B, shows that the final reward and results for GRPO vs. ReSum-GRPO consistently overlap. It would be informative to produce this plots with error bars running this experiment over multiple random seeds to better quantify these claims, otherwise such claims should be weakened.
* Limited novelty / missing citations: The paper’s core idea of compressing conversation context via summarization is an intuitive solution that has been explored by others (, Mem1 (Zhou et al., 2025b) and MemAgent (Yu et al., 2025a)). The authors do cite these, but portray ReSum as “lightweight” in comparison. In truth, the difference is incremental: ReSum outsources summarization to an external model instead of learning memory tokens, but the goal and even the use of RL adaptation are similar. Overall, the contribution feels more like a specific implementation of a growing trend (agentic summarization for long contexts) rather than a pioneering breakthrough. The authors should clarify explicitly what is novel here relative to prior context-management approaches (e.g. is the main novelty just the use of a fixed summary tool and minor RL tweak?). Currently, the work comes across as incremental.
* The “surpassing most open-source web agents” statement in the abstract is too strong without broader, up-to-date comparisons (and with CIs). Please either temper this claim or add head-to-head results versus the strongest recent open agents under matched settings.
* Evaluation relies on an LLM-as-judge signal (also used for RL reward). Please report calibration/human spot-checks or agreement rates, and quantify sensitivity to the chosen judge model.
* Summarizer quality is only evaluated indirectly via downstream Pass@k. Include an intrinsic summary evaluation (coverage of key facts, omission rate) or at least failure analyses where a dropped detail changes the final answer.

**Questions:**

* “Triggers for summarization can be systematic, e.g., exceeding a token budget or reaching a round limit, or agent-initiated, where the policy model decides to summarize for effective context management.”, can the authors provide an ablation on agent-initiated summarization?
* How do you ensure when training the summarizer it will capture the correct long-distance information; and not just learn to compress to keep information/facts helpful for the final evaluated task reward? That is learning to attend and keep information, a form of long horizon credit assignment, information that is vital to a later state, however not immediately necessary for the subsequent states, for example a password to unlock a door at the end of a maze.
* Novelty vs prior art: Can you clarify precisely what is novel in ReSum compared to earlier long-horizon agent frameworks? For example, Mem1 and MemAgent already enable agents to condense context (albeit via learned internal memory) , and concurrent works use similar summarization or memory editing strategies . Which aspect of ReSum’s paradigm or implementation do you consider unique? It would help to explicitly acknowledge overlapping ideas and highlight any distinctive contribution (if any) beyond a “plug-and-play” implementation.
* Please provide error bars (or bootstrap CIs) for all key metrics; and report the number of seeds per configuration. Without this, several improvements could plausibly be within variance.
* How sensitive is performance to the summarization frequency/threshold? An ablation sweeping the trigger threshold (or a learned trigger) would help understand when summarization helps vs. hurts.
* For RL: what fails if one trains with standard GRPO directly under the ReSum prompt format (no “advantage broadcasting”)? A small ablation would justify the need for ReSum-GRPO beyond the inference paradigm itself.

---

> ### Author Response · Authors · 2025-11-19
> **Response to Reviewer SSow (1/3)**
>
> We sincerely thank the reviewer for your thorough and constructive feedback.
>
> ---
>
> > W1 & Q4: Error bars
>
> We thank the reviewer for emphasizing statistical rigor. We follow standard practice in web agent literature by reporting Pass@1 (average success rate across **three independent runs**) and Pass@3 (success within three attempts). Below we provide the standard deviations of Pass@1 across three runs:
>
> WebSailor-3B
> |Setting|GAIA|BrowseComp-zh|BrowseComp-en|
> |-|-|-|-|
> |ReAct|25.6±2.3|8.2±0.7|3.3±1.1|
> |ReSum|35.3±5.7|13.7±1.3|6.8±1.9|
> |GRPO-ReAct|28.5±3.1|11.8±0.6|4.2±0.7|
> |GRPO-ReSum|38.5±3.7|17.3±1.8|8.5±1.3|
> |ReSum-GRPO|37.9±1.7|20.5±0.5|9.2±0.3|
>
> WebSailor-7B
> |Setting|GAIA|BrowseComp-zh|BrowseComp-en|
> |-|-|-|-|
> |ReAct|31.7±1.1|13.2±0.5|5.7±1.1|
> |ReSum|40.5±1.1|17.2±3.4|9.0±0.4|
> |GRPO-ReAct|34.0±0.9|18.7±2.7|5.8±1.1|
> |GRPO-ReSum|37.2±1.4|25.4±0.5|8.5±0.4|
> |ReSum-GRPO|42.4±2.0|27.1±0.5|12.3±0.6|
>
> WebSailor-30B
> |Setting|GAIA|BrowseComp-zh|BrowseComp-en|
> |-|-|-|-|
> |ReAct|45.0±3.9|23.9±2.1|12.8±1.5|
> |ReSum|47.3±1.9|24.1±1.9|16.0±1.7|
> |GRPO-ReAct|48.2±2.0|23.3±0.8|14.3±1.1|
> |GRPO-ReSum|48.5±1.7|29.3±1.1|15.0±1.7|
> |ReSum-GRPO|48.5±3.3|33.3±1.7|18.3±1.5|
>
> The results show **variance <2% in most cases**, demonstrating consistent performance across runs. Given these low variances, the observed improvements, **+4.5% for training-free ReSum and +8.2% for ReSum-GRPO, are statistically meaningful.**
>
> Regarding reproducibility, we have documented all inference configurations (`temperature`, `top_p`, etc.) in **Appendix D.1** to ensure full reproducibility, as the stochasticity in LLM inference is primarily controlled by these parameters.
>
> > W2: Figure 4
>
> Thank you for this careful observation. In Figure 4, each reward point represents the **average correctness** across all samples in a training step (group size=8, batch size=64, yielding **512** samples per step). Below we provide the raw reward values for the last 10 training steps of WebSailor-30B to facilitate quantitative comparison:
> - **GRPO** [0.502, 0.535, 0.568, 0.533, 0.565, 0.609, 0.543, 0.490, 0.619, 0.625]
>      Average reward of last 10 steps: **0.559±0.04**
> - **ReSum-GRPO** [0.672, 0.615, 0.523, 0.623, 0.543, 0.627, 0.582, 0.621, 0.633, 0.656]
>      Average reward of last 10 steps: **0.610±0.04**
>
> ReSum-GRPO achieves **~5% higher average reward** in the final training phase, demonstrating advantages on the training set. Regarding training stability, we follow established practice in resource-intensive web agent RL by conducting a single training run and providing multiple independent inference runs on testing benchmarks. The consistent improvements in final performance (as shown in our response to W1/Q4) confirm the effectiveness of our method.
>
> > W3 & Q3: Limited novelty
>
> We thank the reviewer for this critical question regarding our contributions. We acknowledge the shared high-level goal of context management with prior works like Mem1 [Ref A] and MemAgent [Ref B]. The core novelty of ReSum lies in **introducing a practical paradigm for enabling long-horizon reasoning in pre-existing agents**. The distinction can be framed as **architectural modification vs. paradigmatic enhancement**.
>
> * Mem1/MemAgent innovates through *architectural modification*, introducing learnable memory tokens that require end-to-end training of a new agent from scratch.
>
> * ReSum innovates through a *paradigmatic enhancement*, using a fixed, specialized summary tool externally to manage context for any off-the-shelf ReAct agent. This fundamental difference yields several practical advantages that define ReSum's contribution:
>    * **Training-free utility** ReSum consistently provides gains without any training (Table 1), offering a ready-to-use solution. In contrast, methods like Mem1 can see performance drops in this setting (Table 3 in Section 4.4).
>    * **Data efficiency** When training is desired for specialization, ReSum-GRPO achieves a +8.2% improvement using only **1K samples**, drastically reducing the data cost compared to prior work.
>    * **Forward compatibility** The decoupled summary tool can be independently improved or replaced (e.g., with future, more powerful models), continuously enhancing all ReSum-compatible agents without retraining.
>
> Therefore, ReSum's lightweight and plug-and-play nature is not a minor feature but the essence of its contribution. It addresses a critical need in the community: **enabling long-horizon capabilities for the vast ecosystem of existing agents and for practitioners with limited data or training resources.**
>
> We sincerely appreciate your feedback and have revised the manuscript in the Introduction and Related Works sections to better highlight our contributions.
>
> ---
>
> [Ref A] Zhou et al. *MEM1: Learning to Synergize Memory and Reasoning for Efficient Long-Horizon Agents.* In arXiv preprint, 2025
>
> [Ref B] Yu et al. *MemAgent: Reshaping Long-Context LLM with Multi-Conv RL-based Memory Agent.* In arXiv preprint, 2025

---

> ### Author Response · Authors · 2025-11-19
> **Response to Reviewer SSow (2/3)**
>
> > W4: Surpassing most open-source web agents
>
> Thank you for the careful reading and the chance to clarify our statement. Below we compare against recent **open-weight web agents** on BrowseComp benchmarks under matched settings. For model families with multiple versions, we report the **strongest variant**:
>
> | Model | BrowseComp-zh Pass@1 | BrowseComp-en Pass@1 | Release Date |
> |-------|---------------------|---------------------|--------------|
> | WebSailor-32B | 25.5 | 10.5 | July 3, 2025 |
> | MiroThinker-32B-V0.1 | 17.0 | 13.0 | Aug 8, 2025 |
> | ASearcher-Web-QWQ-32B | 15.6 | 5.2 | Aug 11, 2025 |
> | MiroThinker-32B-V0.2 | 29.4 | 17.2 | Sep 8, 2025 |
> | WebExplorer-8B | 32.0 | 15.7 | Sep 8, 2025 |
> | Tongyi-DeepResearch-30B-A3B | 46.7 | 43.4 | Sep 16, 2025 |
> | **WebSailor-30B + ReSum-GRPO** | **33.3** | **18.3** | - |
>
> Ours ranks 2nd among all open-weight agents at the time of writing. The scores for cited models are from their official reports, which, consistent with standard practice in the field, do not include error bars.
>
>
> > W5: LLM-as-Judge
>
> We appreciate this concern about evaluation rigor and provide our rationale below.
>
> * **Necessity of LLM-as-Judge** Unlike math reasoning where rule-based matching suffices, web agent tasks require semantic understanding due to multilingual expressions, synonym variations (e.g., "passed away in 1890" vs "died in 1890"), and long-form responses. LLM-as-judge provides a reliable and economical solution compared to extensive human annotation, especially for RL training where millions of judgments are required.
>
> * **Judge model selection and validation** We compared four judging models, Qwen2.5-72B-Instruct, GPT-4o-Mini, Qwen3-235B, and Gemini-2.5-Flash, on WebSailor-7B predictions for BrowseComp-zh. The resulting scores were 12.8%, 12.5%, 12.8%, and 12.5%, respectively, showing **variance <0.3%** across all models, demonstrating low sensitivity to judge choice. Manual inspection aligned with Qwen series scoring results. Based on this analysis, we selected Qwen2.5-72B-Instruct for its **reliability and cost-effectiveness**.
>
> * **Alignment with existing literature** Our use of Qwen2.5-72B-Instruct follows standard practice in recent web agent research [Ref C, D],  facilitating direct comparability within this field. Additionally, we maintain consistency with each benchmark's official judging prompt to ensure result reliability.
>
>
> > W6: Evaluation of summarizer
>
> Thank you for this valuable suggestion. We provide evaluation of ReSumTool-30B in **Appendix E.3** with both quantitative results (downstream Pass@k across models) and qualitative analysis (Figure 7 in Appendix E.3), demonstrating that the specialized summary tool effectively collects necessary information. Additionally, Appendix G.3 presents successfully solved trajectories based on summarization, further showing summarizer reliability.
>
> We fully agree that intrinsic evaluation (e.g., coverage, omission rate) would be valuable. However, constructing ground-truth annotations at scale is infeasible: identifying essential facts for each open-ended query requires backward reasoning from answers through iterative search and verification with high accuracy standards, often necessitating human annotation. We view this as important future work for the community.
>
> ---
>
> [Ref C] Gao et al. *Beyond Ten Turns: Unlocking Long-Horizon Agentic Search with Large-Scale Asynchronous RL.* In arXiv preprint, 2025
>
> [Ref D] Dong et al. *Agentic Reinforced Policy Optimization.* In arXiv preprint, 2025

---

> ### Author Response · Authors · 2025-11-19
> **Response to Reviewer SSow (3/3)**
>
> > Q1 & Q5: Ablation for summarization trigger
>
> We thank the reviewer for these insightful questions regarding trigger mechanisms.
>
> For agent-initiated summarization, our preliminary experiments revealed that enabling agents to autonomously manage their context currently **exceeds their capabilities**. Effective implementation would require not only specific prompting strategies but also cold-start trajectories to familiarize agents with this paradigm. We therefore consider learnable triggers as an exciting direction for future work.
>
> For summarization trigger frequency, we conducted the suggested sensitivity study across multiple fixed thresholds (8K, 16K, 24K, and 32K tokens). Using WebSailor-7B in a training-free setting with consistent tool call budgets, we obtained the following results and findings:
>
> |Summary Trigger|Pass@1 GAIA (%)|# Summary|# Token (K)|# Tool|
> |---------------|---------------|------------|-----------|------|
> |ReAct|31.7|0|17.4|10.3|
> |8K|44.5|2.6|36.0|17.2|
> |16K|40.8|1.1|29.4|13.6|
> |24K|39.8|0.4|20.7|11.2|
> |32K|40.5|0.3|21.8|12.9|
>
> |Summary Trigger|Pass@1 BrowseComp-zh (%)|# Summary|# Token (K)|# Tool|
> |---------------|----------------|------------|-----------|------|
> |ReAct|13.2|0|29.2|10.7|
> |8K|18.8|5.5|84.5|33.6|
> |16K|21.1|3.7|60.5|21.1|
> |24K|21.5|2.2|58.4|20.4|
> |32K|17.2|1.8|50.6|17.6|
>
> * **Consistent benefits of ReSum** All trigger frequencies outperform ReAct, confirming that summarization provides gains through both extended exploration and evidence distillation.
>
> * **Dataset-specific optimal frequency** Simpler benchmarks like GAIA benefit from frequent summarization, e.g., 8K, while more challenging benchmarks like BrowseComp-zh require longer uninterrupted chains as optimal at 24K.
>
> * **Cost-performance trade-off** More frequent summarization significantly increases computational costs due to additional summary invocations, higher token consumption, and an increased number of tool calls. We empirically found that under frequent triggers, e.g., 8K, providing fresh summaries more often stimulates the agent to engage in more persistent verification and exploratory search. Thus, our current implementation, based on the **actual context limit of 32K**, was chosen to **balance between performance gains and practical efficiency**.
>
>
> > Q2: Summary tool quality control
>
> Thank you for this insightful question. We acknowledge that we did not explicitly address long-horizon credit assignment in summarizer training. Our quality control of the summary tool (**Appendix E**) relies on powerful teacher model selection, abundant high-quality data curation, and rigorous case analysis to ensure reliability.
>
> We fully agree that guiding the summarizer to retain long-range dependency information warrants further investigation. As part of our future work, we plan to explore more sophisticated frameworks, e.g., incorporating an LLM-as-Judge during rollouts to evaluate whether summaries effectively preserve critical long-horizon clues. Achieving this would require large-scale rollout data, a carefully selected judge model, and meticulously designed prompts. We respectfully consider this a meaningful yet separate research endeavor.
>
> We will include a discussion of this limitation in the revised manuscript. Thank you again for your valuable feedback.
>
>
> > Q6: Ablation on advantage broadcasting
>
> We appreciate this question regarding the necessity of advantage broadcasting. A direct ablation of standard GRPO for ReSum is technically infeasible for fundamental reasons:
>
> * **Context limit** Complete ReSum trajectories with multiple segments exceed the 32K context window, making standard GRPO processing physically impossible.
>
> * **Data utilization** Without advantage broadcasting, pre-summary trajectory segments become unusable for training as they lack proper reward signals, discarding most collected segments.
>
> Given these constraints, advantage broadcasting is not an optimization but a necessary component that enables effective learning from all segments of long-horizon tasks. The substantial resource requirements for RL training further precluded extensive ablation studies on this intuitively problematic variant.
>
> ---
>
> *Once again, we sincerely thank you for your meticulous reading, constructive feedback, and valuable suggestions. We hope the additional experiments and clarifications in the revised manuscript have addressed the points you raised. Please let us know if there are any further questions we can answer.*

---

> ### Author Response · Authors · 2025-11-24
> **Thank you & Looking forward to further discussion**
>
> Dear Reviewer SSow,
>
> We extend our highest respect for the time and effort you devoted to reviewing our manuscript, we can truly sense the considerable attention you have given to every detail. To facilitate better communication and express our gratitude, we sincerely summarize our rebuttal as follows:
>
>
> * **Novelty & contribution** `Weakness3, Question3` We respectfully clarified the fundamental distinction between MEM1/MemAgent and our work as **architectural modification vs. paradigmatic enhancement**. ReSum introduces a practical paradigm for enabling long-horizon reasoning in pre-existing agents, offering advantages including *training-free utility*, *data-efficient adaptation*, and *forward compatibility*.
>
> * **Performance variances** `Weakness1, Question4` We have supplemented results with standard deviations from three independent runs. Variances are below 2% in most cases, confirming that the improvements from ReSum (+4.5%) and ReSum-GRPO (+8.2%) are **statistically meaningful**.
>
> * **Ablation on summarization trigger** `Question1, Question5` We have supplemented an analysis of trigger frequency sensitivity, which shows the benefits of ReSum across different settings.
>
> * **LLM-as-Judge** `Weakness5` We respectfully justified this choice based on its economic necessity (particularly for RL requiring millions of judgments), semantic flexibility (handling diverse expressions beyond rule-based matching), minimal variance across judge models, alignment with established practice, and strict adherence to official evaluation prompts.
>
> For other points such as training dynamics (`Weakness2`) and comparisons with open-source agents (`Weakness4`), we have provided additional statistical support and revised the manuscript accordingly for improved clarity. We sincerely thank you for your invaluable suggestions. For any other issues not mentioned here, please refer to our detailed rebuttal response.
>
> We sincerely and humbly believe we have responded to your concerns, and would be truly grateful if you could let us know **whether our responses have adequately addressed your concerns**. Thank you again for your time and insightful comments.
>
> Warm regards,
>
> Authors of submission 8387

---

> ### Author Response · Authors · 2025-11-26
> **[Only 1 Week Remaining] A Gentle Reminder for Feedback**
>
> Dear Reviewer SSow,
>
> We sincerely thank you for your valuable feedback and the time you have invested in reviewing our submission. We have carefully addressed the concerns you raised, incorporated relevant discussions and experiments, and revised the manuscript accordingly.
>
> As the deadline for discussion is approaching, we kindly request your feedback to confirm **whether our responses sufficiently address your concerns**. If there are any remaining issues, we would greatly appreciate the opportunity to address them to further improve the quality of our work.
>
> Thank you again for your time and consideration.
>
> Warm regards,
>
> Authors of submission 8387

---

### Author Response · Authors · 2025-12-02
**Global Response**

Dear Reviewers and ACs,

We would like to express our deepest gratitude for your diligent coordination in the review process and for the insightful feedback provided by the reviewers. During the author-reviewer discussion phase, we endeavored to address the concerns raised. As most reviewers did not engage in the discussion phase, we would like to clarify the most critical concerns regarding our work.

|Dimension|Key Concern|Our Response|
|-|-|-|
|Contribution|Comparison with existing works MEM1/MemAgent (`Reviewer SSow`)|The core distinction is **architectural modification vs. paradigmatic enhancement**. MEM1/MemAgent modify the agent architecture, requiring extensive end-to-end training for adaptation. ReSum instead introduces a lightweight paradigm that seamlessly integrates with off-the-shelf ReAct agent. Its practical advantages include in **training-free usage**, **high data efficiency**, and **forward compatibility**.|
|Contribution|Comparison with world-models-augmented agent (`Reviewer Je2p`)|ReSum and world models address different problems: **context window exhaustion vs. action selection under uncertainty**. Their roles are orthogonal: world models act as dense, forward-looking planners invoked frequently per step, while ReSum’s summary tool serves as a sparse, on-demand, backward-looking context manager.|
|Efficiency & Practicality|Comparison with leading pre-trained models (`Reviewer Je2p`)|The supplemented cost-performance analysis highlights that leading LLMs are highly resource-intensive. In contrast, ReSum offers **a compelling alternative by boosting the performance of smaller, more deployable base agents** under tight hardware constraints, making it a practical solution for on-device or cost-sensitive applications.|
|Efficiency & Practicality|Orthogonal effectiveness to stronger models with larger context windows (`Reviewer Je2p, FmG8`)|Our supplementary experiments (Appendix F) on the SOTA Tongyi-DeepResearch agent across five context limits **consistently demonstrate ReSum's improvement over ReAct**, showing gains are **orthogonal to the base agent's capabilities and context capacity**.|
|Efficiency & Practicality|Inference latency introduced by ReSum (`Reviewer oXox`)|The added latency is **modest and justified**. ReSum introduces an average of 1–3 additional calls per trajectory, which is a modest overhead compared to the total trajectory (10+ tool calls). This cost enables the resolution of otherwise impossible queries, representing a cost-performance trade-off.|
|Framework Robustness|Sensitivity to summarization trigger frequency (`Reviewer oXox, SSow`)|Ablation studies on trigger frequency reveal that all tested frequencies outperform ReAct, with the optimal setting being dataset-specific. Our chosen trigger, aligned with the actual 32K context limit, achieves a balance between cost and effectiveness.|
|Framework Robustness|Quality control of ReSumTool-30B (`Reviewer FmG8, oXox, SSow`)|We implemented **rigorous quality control (Appendix E)**, including careful teacher model selection, abundant high-quality data curation, and detailed case analysis. The distilled model's effectiveness is empirically shown in Table 1.|

We respectfully state that **some concerns challenge the common practice in contemporary web agent research** and **may not fully warrant a negative assessment.**
* **LLM-as-Judge** Exact string matching is inadequate for evaluating open-ended question answering. Therefore, we use Qwen2.5-72B-Instruct as the judging model, which aligns with established practices. Additionally, our comparisons demonstrate minimal variance across judging models and strong agreement with human evaluations, ensuring reliable and consistent assessment.
* **Model variances** We have supplemented results with variance across three runs (<2% in most cases), showing that the improvements brought by ReSum (+4.5) and ReSum-GRPO (+8.2) are statistically significant and meaningful.

For the paper writing and presentation, we carefully revised the manuscript to incorporate new experiments, clarify the methodology, highlight key contributions, and address ambiguity and confusion, as suggested by the reviewers. We sincerely thank the reviewers for their valuable feedback.

Finally, we would like to re-emphasize our contributions. Web agents are becoming increasingly popular but remain fundamentally constrained by the context limits of modern LLMs. Given the high cost of LLM post-training, our work **pioneers a lightweight and practical paradigm to equip existing agents with long-horizon intelligence**. Its training-free utility, data efficiency for paradigm adaptation, and orthogonal effectiveness represent a meaningful step forward in this field. We sincerely hope our contributions are appreciated. We genuinely hope that our response will assist the ACs in rendering a comprehensive decision. Many thanks for your time.

Best,

Authors of Paper #8387

---

### Meta-Review · Area_Chair_C59q · 2026-01-07

**Summary:**

The paper appears incremental in nature. Reviewer SSow (Weakness 3 / Question 3) raises concerns about the level of novelty, and Question 2 points to a potential limitation in that the summarizer may not adequately capture long-distance information. Additional factors affecting the decision include limited justification for the use of LLM-as-a-judge (Reviewer FmG8) and an unclear response to Weakness 1 / Question 4 noted by Reviewer SSow.

**Reviewer Concerns:**

The authors’ responses address several of the reviewers’ concerns; however, a number of issues remain unresolved. In particular:

- Reviewer SSow, Question 2: The concern regarding whether the summarizer can reliably capture long-distance information is acknowledged by the authors but deferred to future work, leaving the limitation unaddressed in the current submission.

- Reviewer FmG8, Weakness 2.1: The justification for using an LLM-as-a-judge remains insufficient. The explanation provided in the rebuttal does not convincingly address the reviewer’s concern.

- Reviewer FmG8, Weakness 2.3: The reviewer suggested evaluating GRPO without recent history as a baseline, which, under the authors’ own rationale, should be relatively weak. This experiment was not conducted, leaving the empirical evaluation incomplete.

- Reviewer SSow, Weakness 1 / Question 4: The authors’ response is unclear and does not adequately resolve the concern.

Overall, while the paper makes some progress, it appears relatively incremental in nature and falls below the bar for acceptance at ICLR.

**Reviewer Scores:**

It is very likely that Reviewer FmG8 will keep the score of 4. For the rest of the three
reviewers, it is all possible that they increase the score from 4 to 6, depending on their
leniency. Reviewer Je2p is the most likely to increase the score.

**Reviewer SSow**
- Weakness 1 / Question 4: Random seeds + error bars to see if the improvement
is consistent.

This concern is largely addressed. However, some confusion remains regarding the reporting of both GRPO-ReSum and ReSum-GRPO. In particular, GRPO-ReSum is not clearly defined or discussed in the paper, and its reported results differ from those of ReSum-GRPO, with the two variants often exhibiting statistically significant differences. Clarification of this distinction is needed.

- Weakness 2: ReSum-GRPO in Fig. 4 b) is over-claiming.

Somewhat addressed. The authors' response did not directly answer the reviewer’s
request, but the authors’ rationale was reasonable.

- Weakness 3 / Question 3: Limited novelty, missing citations.

Addressed, though the work is not a pioneering
breakthrough.

- Weakness 4: The claim that “surpassing most open-source web agents” is too
strong.

Addressed.

- Weakness 5: Evaluation relies on LLM as judge.

Addressed.

- Weakness 6: Summarizer quality not intrinsically evaluated.

Addressed. The authors provided a qualitative analysis and explained infeasibility of
quantitative measures.

- Question 1: Ablation and hyperparameter sensitivity for summarization.

Addressed.

- Question 2: Ensure summarizer captures correct long horizon info.

Unaddressed and stated as future work.

- Question 6: Ablation on advantage broadcasting.

Addressed; the authors explained infeasibility.

**Reviewer oXox**
- Weakness 1 / Question 1: The triggering strategy is suboptimal and needs
more testing.

Provide some rationale and results. Q1 is addressed; whether W1 is addressed or not
depends on the reviewer’s belief.

- Weakness 2 / Question 2: Risk of error propagation.

Addressed.

- Weakness 3: Inference latency of tool calling.

Mostly addressed, though the exact latency of each tool calling was not discussed.

- Weakness 4: Training data.

Addressed.

**Reviewer FmG8**

- Weakness 1: Unsubstantiated or inaccurate claims.

Addressed.

- Weakness 2.1: Justification for LLM-as-a-judge.

This concern remains unaddressed. The authors did not provide sufficient evidence that aligning the evaluation model offers advantages over the benchmark’s existing evaluation, particularly in the absence of evidence that the latter is inaccurate. This point is also reiterated by the reviewer in the follow-up.

- Weakness 2.2: Choice of baselines.

Addressed.

- Weakness 2.3: GRPO without recent history as baseline.

Unaddressed. If the authors’ rationale is true, this will be a weak baseline and the
authors should report.

- Weakness 2.4: Experiments on stronger models.

Addressed.

- Weakness 3: Contradicting statements.

Addressed.

- Question 1: Summarization quality.

Addressed. The authors provided a qualitative analysis and explained infeasibility of
quantitative measures.

- Question 2: Loss computation.

Addressed.

- Question 3: Model variance.

Same as SSow point 1.

**Reviewer Je2p**

All comments were well-addressed.

---

### Decision · Program_Chairs · 2026-01-26

Reject